# Management of Patients with Newly Diagnosed Desmoid Tumors in a First-Line Setting

**DOI:** 10.3390/cancers14163907

**Published:** 2022-08-12

**Authors:** Stefano Testa, Nam Q. Bui, Gregory W. Charville, Raffi S. Avedian, Robert Steffner, Pejman Ghanouni, David G. Mohler, Kristen N. Ganjoo

**Affiliations:** 1Department of Medicine, Stanford University, Stanford, CA 94304, USA; 2Division of Oncology, Department of Medicine, Stanford University, Stanford, CA 94304, USA; 3Department of Pathology, Stanford University, Stanford, CA 94304, USA; 4Department of Orthopedic Surgery, Stanford University, Redwood City, CA 94063, USA; 5Department of Radiology, Stanford University, Stanford, CA 94304, USA

**Keywords:** desmoid tumor, active surveillance, surgery, hormonal therapy, tyrosine kinase inhibitors, cryoablation, high intensity focused ultrasound, chemotherapy

## Abstract

**Simple Summary:**

Desmoid tumors are benign neoplasms that invade locally, causing significant disability and morbidity. Historically, patients with desmoid tumors have been treated with surgery despite the significant morbidity associated with this modality. Less invasive treatments have emerged, including active surveillance, systemic therapy, radiotherapy, and local ablation. However, it remains unclear which patients would benefit most from an initial conservative rather than interventional approach. To answer this question, we retrospectively analyzed 262 patients with desmoid tumors treated at our institution over a period of 30 years. Our results suggest that initial active surveillance is a good option for patients with small and minimally symptomatic desmoid tumors, while tyrosine kinase inhibitors, local ablation, and surgery seem to be equally effective in those with more aggressive disease.

**Abstract:**

The initial management of desmoid tumors (DTs) is shifting from surgery towards active surveillance, with systemic and locally ablative treatments reserved for enlarging and/or symptomatic disease. However, it remains unclear which patients would benefit most from an initial conservative rather than interventional approach. To answer this question, we retrospectively analyzed adult and pediatric patients with DTs treated at a tertiary academic cancer center between 1992 and 2022. Outcomes measured were progression-free survival (PFS) and time to next treatment (TTNT) after first-line therapy. A total of 262 treatment-naïve patients were eligible for analysis with a median age of 36.5 years (range, 0–87 years). The 5-year PFS and the median TTNT (months) after first-line treatment were, respectively: 50.6% and 69.1 mo for surgery; 64.9% and 149.5 mo for surgery plus adjuvant radiotherapy; 57.1% and 44.7 mo for surgery plus adjuvant systemic therapy; 24.9% and 4.4 mo for chemotherapy; 26.7% and 5.3 mo for hormonal therapy; 41.3% and 29.6 mo for tyrosine kinase inhibitors (TKIs); 44.4% and 8.9 mo for cryoablation and high intensity focused ultrasound; and 43.1% and 32.7 mo for active surveillance. Age ≤ 40 years (*p* < 0.001), DTs involving the extremities (*p* < 0.001), a maximum tumor diameter > 60 mm (*p* = 0.04), and hormonal therapy (*p* = 0.03) predicted a higher risk of progression. Overall, our results suggest that active surveillance should be considered initially for patients with smaller asymptomatic DTs, while upfront TKIs, local ablation, and surgery achieve similar outcomes in those with more aggressive disease.

## 1. Introduction

Desmoid tumors (DTs), also known as desmoid fibromatoses, are locally invasive mesenchymal neoplasms with no potential for metastatic spread. These are rare tumors accounting for less than 3% of soft tissue neoplasms [1,2]. Approximately 80% of DTs develop sporadically, with the remainder arising in the context of familial adenomatous polyposis (FAP) [3]. The pathogenesis of DTs involves the activation of the β-catenin/WNT pathway, with mutations in the *CTNNB1* gene accounting for approximately 80% of DTs, mostly sporadic cases, and *APC* mutations accounting for around 20% of cases, particularly in DTs associated with FAP [4,5]. 

The clinical behavior of DTs is quite variable, with some tumors showing high recurrence rates after surgery and others that regress spontaneously without treatment [6,7]. Surgery has historically been the mainstay of treatment for patients with DTs, even though this is often technically difficult, associated with a high degree of morbidity and disability, and characterized by a high recurrence rate, even when negative margins are achieved [8,9]. For these reasons, active surveillance is currently the preferred first-line approach for patients with minimally symptomatic DTs [10,11]. This approach consists of periodic imaging monitoring of tumor growth with the option to pursue treatment in case of tumor progression or worsening symptoms. Systemic therapy has been used for patients who fail an active surveillance approach [12]. Among systemic therapies, there are anti-estrogenic agents (tamoxifen, toremifene, leuprolide), cytotoxic chemotherapy (methotrexate, vinblastine, doxorubicin), tyrosine kinase inhibitors (sorafenib, imatinib, pazopanib) and non-steroidal anti-inflammatory drugs (celecoxib, sulindac, meloxicam). Radiotherapy is also a valid alternative for some patients and has shown good rates of local control with good progression-free survival (PFS) intervals, even though it brings a small but significant risk of developing secondary malignancies [13,14]. Lastly, new approaches for local ablation have emerged recently, such as cryoablation and magnetic resonance-guided high intensity focused ultrasound (MR-HIFU), which are usually reserved for symptomatic and/or enlarging DTs [15,16]. 

It remains unclear which patients would benefit most from an upfront surgical approach rather than from initial non-operative management involving either medical treatment, locally ablative therapies, or active surveillance. Here, we retrospectively analyzed both adult and pediatric patients with desmoid tumors treated at a tertiary academic cancer center over a period of 30 years. We compared outcomes between patients treated with a primary surgical modality and patients managed with either active surveillance or with a non-surgical approach including medical therapy or local treatments. At the same time, we correlated patient and tumor characteristics with outcomes to identify features with prognostic value. 

## 2. Materials and Methods

### 2.1. Patient Selection

A total of 262 treatment-naïve patients with a pathologically confirmed diagnosis of desmoid tumor (DT) were included in the study. Patients were selected from a database of 480 patients treated at a single tertiary academic cancer center between 1992 and 2022. Of these, 211 patients were excluded because of incomplete records or because they received most of their treatment at a different institution. Seven patients were excluded due to the presence of a concomitant different malignancy.

### 2.2. Active Surveillance and Treatment

At our institution, active surveillance consisted of clinical visits, including history and physical examination together with either CT or MRI imaging, performed every three months during the first year after diagnosis, every six months during the second year, and once a year for the following years until disease progression. For the other treatment groups, including surgery, medical treatments, and local ablation, only patients who received these treatment modalities as first-line, without undergoing active surveillance beforehand, were included. The decision to pursue treatment rather than active surveillance was based on the presence of symptoms and proximity to vital structures or organs with high risk of disability and morbidity in case of tumor progression. 

### 2.3. Assessment of Adverse Events and Complications

Adverse events and complications after the first-line treatment were assessed through clinical visits including a history and physical examination. Clinical visits were performed every three months for the first year after treatment, every six months for the second year and then yearly for the subsequent years. Adverse reactions were categorized with the Common Terminology Criteria for Adverse Events (CTCAE) version 5.0 whenever possible [17].

### 2.4. Statistical Analysis

Progression-free survival (PFS) was measured from time of diagnosis to the time of first clinical disease progression or was censored at the last follow-up. Clinical disease progression was determined based on a review of the treating physician’s notes and corresponded to either worsening symptoms or tumor enlargement during follow-up imaging. Time to next treatment (TTNT) was measured from the date of initiation of the first-line treatment to the date of initiation of the second-line treatment, or was censored at the last follow-up. Occasionally, patients who received MR-HIFU or cryoablation needed to receive multiple sequential treatments in a staged approach. In these cases, the second repeat treatment was still considered in the calculation of the TTNT, resulting in underestimation of the TTNT. The Kaplan–Meier method was used to calculate the PFS and the TTNT. The Log-rank method was used to measure differences between the survival curves. Survival analysis was performed in R through the packages survminer (https://cran.r-project.org/package=survminer, (accessed on 1 March 2022)) and ggplot2 [18,19]. The Cox proportional hazards analysis was used to calculate the Hazard Ratio (HR) and was performed in SPSS version 28 (IBM). For descriptive statistics, continuous variables were characterized using the median and the range of the distribution while categorical variables were described through proportions. Differences between proportions were measured through the Chi-square test which was performed in GraphPad Prism version 9.3.1 (GraphPad Software, San Diego, CA, USA).

## 3. Results

### 3.1. Demographic and Clinical Patient Characteristics 

The baseline features and treatment information of the patients in the study are summarized in Table 1. The median age at diagnosis was 36.5 years (range, 0–87 years) with the extremities involved in most cases (*n* = 85, 32%). Most patients were female (*n* = 185, 70.6%). The most common first-line treatment was surgery (*n* = 134, 51.1%), followed by surgery plus adjuvant radiotherapy (*n* = 27, 10.3%). Among non-surgical treatments, local therapies including either cryoablation or MR-HIFU (*n* = 20, 7.6%), and tyrosine kinase inhibitors (TKIs, *n* = 21, 8.0%) were the most common approaches. Further details about the frequency of each specific first-line treatment given can be found in Appendix A. 

Observing treatment patterns over time (Appendix A), we found that systemic treatments were more commonly used as first-line between 2008 and 2022 compared to 1992 and 2007 (22.6% vs. 4.4%, *p* = 0.005), while surgery with adjuvant radiotherapy was most common between 1992 and 2007 than 2008 and 2022 (33.3% vs. 5.5%, *p* < 0.0001). No patients were managed with active surveillance between 1992 and 2007, while between 2008 and 2022, there were 19 patients (8.8%) that were initially managed expectantly (*p* = 0.03).

Genetic testing with targeted next-generation sequencing was available at diagnosis for 45 patients (17.2%), with the most common mutation identified being the missense mutation p.T41A in *CTNNB1* (*n* = 20, 7.6%). Fourteen patients in the study had a diagnosis of FAP (5.3%). Median follow-up was 63 months (range, 1–286 months). 

### 3.2. Progression-Free Survival

The only first-line treatment associated with reduced risk of progression was surgery plus radiotherapy (HR:0.4, 95%CI: 0.2–0.9, *p* = 0.02, Table 2). In contrast, receiving upfront hormonal therapy correlated with increased risk of progression (HR:2.2, 95%CI: 1.1–4.5, *p* = 0.03). There was no difference in 5-year PFS between patients treated with upfront surgery alone (50.6%, 95%CI: 41–59.5%, *n* = 130), surgery plus radiotherapy (64.9%, 95%CI: 43.2–80%, *n* = 26) or surgery plus systemic therapy (57.1%, 95%CI: 17.2–83.7%, *n* = 9, Appendix A).

Patients treated with first-line chemotherapy had worse 5-year PFS (24.9%, 95%CI: 4.3–54.2%, *n* = 13) compared to those treated with upfront surgery alone (*p* = 0.006) or surgery plus radiotherapy (*p* = 0.003), but similar 5-year PFS compared to those treated with surgery plus systemic therapy (*p* = 0.14, Figure 1A). Patients who received either cryoablation or MR-HIFU as first-line treatment had similar 5-year PFS (44.4%, 95%CI: 16.8–69.1%, *n* = 20) compared to those treated with surgery (*p* = 0.70) or surgery plus systemic therapy (*p* = 0.33), but worse 5-year PFS than patients initially treated with surgery plus radiotherapy (*p* = 0.03, Figure 1B). There was also no significant difference in 5-year PFS between patients treated with first-line TKIs (41.3%, 95%CI: 13.7–67.4%, *n* = 21) and those treated with surgery (*p* = 0.78), surgery plus radiotherapy (*p* = 0.15), or surgery plus systemic therapy (*p* = 0.80, Figure 1C). Patients treated with first-line hormonal therapy had lower 5-year PFS (26.7%, 95%CI: 4.7–56.3%, *n* = 10) than those treated with upfront surgery alone (*p* = 0.02) or surgery plus radiotherapy (*p* = 0.002) but no different 5-year PFS when compared to those receiving surgery plus systemic therapy (*p* = 0.17, Figure 1D). 

There was no difference in 5-year PFS between patients treated with methotrexate-vinblastine and those who received a different chemotherapy regimen (Appendix A, Appendix A), as well as between those treated with tamoxifen with or without NSAIDs and leuprolide with or without NSAIDs (Appendix A). There was instead a non-significant trend for better 5-year PFS in patients treated with imatinib with or without radiotherapy compared to those who received sorafenib (Appendix A), and in those treated with MR-HIU compared to cryoablation (Appendix A). 

Patients with tumors involving the extremities had worse 5-year PFS (30.4%, 95%CI: 20.2–41.2%, *n* = 85) than patients with abdominal wall (66.2%, 95%CI: 49.4–78.6%, *n* = 50, *p* = 0.0003), or intra-abdominal desmoid tumors (68.1%, 95%CI: 52.2–79.7%, *n* = 59, *p* = 0.0001, Figure 2A). There was no difference in 5-year PFS between patients with DTs of the chest wall and those with head and neck DTs (Appendix A, Appendix A). Patients who were 40 years-old or younger at diagnosis had worse 5-year PFS (37.7%, 95%CI: 28.8–46.5%, *n* = 151) than those older than 40 years (64.1%, 95%CI: 53.1–73.2%, *n* = 111, *p* = 0.0004, Figure 2B). Patients with a maximum tumor diameter greater than 60 mm at diagnosis had worse 5-year PFS (42.1%, 95%CI: 32.6–51.3%, *n* = 139) compared to those with a tumor diameter equal or less than 60 mm (57.2%, 95%CI: 46.7–66.4%, *n* = 123, *p* = 0.03, Figure 2C).

### 3.3. Time to Next Treatment

TTNT was significantly longer for patients who received first-line surgery plus radiotherapy (HR: 0.4, 95%CI: 0.2–0.7, *p =* 0.005) or surgery alone (HR: 0.6, 95%CI: 0.4–0.9, *p* = 0.008, Table 3). TTNT was instead shorter for patients treated with first-line hormonal therapy (HR: 3.3, 95%CI: 1.7–6.6, *p* < 0.001), chemotherapy (HR: 4.9, 95%CI: 2.8–9.2, *p* < 0.001), and for those that received cryoablation or MR-HIFU (HR: 2.4, 95%CI: 1.3–4.4, *p* = 0.004), although this reflects, in part, the division of planned staged ablation sessions into separate treatments.

There was no difference in median TTNT between patients treated initially with surgery (69.1 months, 95%CI: not reached, *n* = 130), surgery plus radiotherapy (149.5 months, 95%CI: not reached, *n* = 26), or surgery plus systemic therapy (44.7 months, 95%CI: 2.9–86.4 months, *n* = 9, Appendix A).

Patients treated with upfront chemotherapy had shorter median TTNT (4.4 months, 95%CI: 2.0–6.7 months, *n* = 13) compared to those treated initially with surgery alone (*p* < 0.0001), surgery plus radiotherapy (*p* < 0.0001), or surgery plus systemic therapy (*p* = 0.009, Figure 3A). Patients treated with cryoablation or MR-HIFU as first-line had similar median TTNT (8.9 months, 2.7–15.1 months, *n* = 20) compared to those treated with initial surgery plus systemic therapy (*p* = 0.15), but shorter median TTNT than those treated with initial surgery plus radiotherapy (*p* < 0.0001) or surgery alone (*p* = 0.0004, Figure 3B). Patients treated with TKIs upfront had shorter median TTNT (29.6 months, 95%CI: 8.0–51.1 months, *n* = 21) than those treated with initial surgery plus radiotherapy (*p* = 0.006), but similar median TTNT when compared to patients treated with surgery alone (*p* = 0.18) or surgery plus systemic therapy (*p* = 0.62, Figure 3C). First-line hormonal therapy resulted in shorter median TTNT (5.3 months, 95%CI: 0.0–13.6, *n* = 10) when compared to surgery alone (*p* < 0.0001) or surgery plus radiotherapy (*p* < 0.0001) with no difference in median TTNT when compared to surgery plus systemic therapy (*p* = 0.05, Figure 3D). 

The median TTNT was similar between patients treated with methotrexate-vinblastine compared to other chemotherapy regimens (Appendix A, Appendix A), as well as between those treated with tamoxifen with or without NSAIDs and leuprolide with or without NSAIDs (Appendix A), and between those who received MR-HIFU compared to cryoablation (Appendix A). Additionally, there was a non-significant trend for worse median TTNT in patients treated with sorafenib compared to those who received imatinib with or without radiotherapy (Appendix A). 

Patients with DTs involving the extremities had worse median TTNT (19.6 months, 95%CI: 10.4–28.8 months, *n* = 85 Figure 4A) compared to those with intra-abdominal DTs (146.7 months, 95%CI: not reached, *n* = 59, *p* = 0.001) and those with DTs of the abdominal wall (170.1 months, 95%CI: not reached, *n* = 50, *p* = 0.001). There was no difference in median TTNT between patients with DTs of the chest wall and those with DTs of the head and neck (Appendix A, Appendix A). Patients 40 years old or younger at diagnosis had worse median TTNT (27.5 months, 95%CI: 19.2–35.7 months, *n* = 151) than those older than 40 years (110.5 months, 95%CI: 25.3–195.6 months, *n* = 111, *p* = 0.01, Figure 4B). There was a non-significant trend for worse median TTNT for patients with DTs larger than 60 mm (32.7 months, 95%CI: 17.7–47.8, *n* = 139) compared to those with DTs that were 60 mm or smaller (69.1 months, 95%CI: 0.0–163.8, *n* = 123, *p* = 0.05, Figure 4C). 

In the multivariate analysis, first-line surgery plus radiotherapy correlated with longer TTNT (HR: 0.3, 95%CI: 0.1–0.7, *p* = 0.004), while receiving upfront chemotherapy correlated with shorter TTNT (HR: 4.0, 95%CI: 1.7–9.6, *p* = 0.002). 

### 3.4. Active Surveillance vs. Upfront Treatment 

Patients who initially underwent active surveillance had similar 5-year PFS (43.1%, 95%CI: 12.2–71.4%, *n* = 19) compared to those treated with first-line surgery (*p* = 0.65), or surgery plus systemic therapy (*p* = 0.33), but worse 5-year PFS compared to those who received initial surgery plus radiotherapy (*p* = 0.03, Figure 5A). Similarly, patients managed expectantly had shorter median TTNT (32.7 months, 95%CI: 22.4–42.9) when compared to those who received initial surgery plus radiotherapy (*p* = 0.03), but similar median TTNT compared to those who received surgery alone (*p* = 0.58), or surgery plus systemic therapy as first-line (*p* = 0.76, Figure 5B). 

Additionally, there was no difference in 5-year PFS between patients who were initially managed with active surveillance and those who received either chemotherapy (*p* = 0.49), TKIs (*p* = 0.28), hormonal therapy (*p* = 0.31) or cryoablation/MR-HIFU as first-line (*p* = 0.78, Figure 5C). In contrast, patients who were initially managed expectantly had longer median TTNT than those who received first-line chemotherapy (*p* = 0.0006), cryoablation/MR-HIFU (*p* = 0.03), or hormonal therapy (*p* = 0.004), but similar median TTNT compared to those who received TKIs (*p* = 0.47, Figure 5D). 

### 3.5. Baseline Characteristics and Treatment Modality

Baseline patient and tumor characteristics can influence the first-line treatment chosen and, as we have observed, some of these features can correlate with outcomes. To better assess the impact of baseline patient characteristics on treatment outcomes and prognosis, we compared tumor size, age at diagnosis and tumor location across treatment groups (Figure 6). 

As far as tumor location, patients who received surgery had a lower proportion of extremity DTs compared to those who received surgery plus adjuvant radiotherapy (22% vs. 63%, *p* < 0.0001, Figure 6A), cryoablation/MR-HIFU (22% vs. 40%, *p* = 0.005), TKIs (22% vs. 43%, *p* = 0.001), hormonal therapy (22% vs. 60%, *p* < 0.0001), or chemotherapy (22% vs. 46%, *p* = 0.0003). Patients managed with active surveillance had a similar proportion of extremity DTs to those treated with surgery (32% vs. 22%, *p* = 0.11) or TKIs (32% vs. 22%, *p* = 0.10), but lower compared to patients treated with surgery plus adjuvant radiotherapy (32% vs. 63%, *p* < 0.0001), hormonal therapy (32% vs. 60%, *p* < 0.0001) or chemotherapy (32% vs. 46%, *p* = 0.04). A more detailed description of tumor location in each treatment group is shown in Appendix A.

In terms of age at diagnosis, the proportion of patients 40 years-old or younger was similar between those treated with surgery and surgery plus radiotherapy (53% vs. 63%, *p* = 0.15), but higher in those who received surgery with adjuvant systemic therapy compared to surgery alone (67% vs. 53%, *p* = 0.04, Figure 6B). Patients treated with TKIs were more often older than 40 years-old than those in any other treatment group. Patients treated with hormonal therapy and those treated with TKIs had a similar proportion of patients younger than 40 years-old (85% vs. 80%, *p* = 0.35), which was higher than in any other treatment group. Patients managed with active surveillance had a similar proportion of patients 40 years-old or younger to those treated with surgery (53% vs. 53%, *p* > 0.99), surgery plus adjuvant radiotherapy (53% vs. 63%, *p* = 0.15) and cryoablation or MR-HIFU (53% vs. 60%, *p* = 0.31).

Regarding tumor size at diagnosis, patients treated with surgery were less likely to have tumors greater than 60 mm compared to those treated with TKIs (45% vs. 67%, *p* = 0.001), hormonal therapy (45% vs. 70%, *p* = 0.0003) or chemotherapy (45% vs. 85%, *p* < 0.0001, Figure 6C). Additionally, the proportion of patients with tumors greater than 60 mm was lower among those treated with surgery compared to those who received cryoablation/MR-HIFU (45% vs. 70%, *p* = 0.0003), or surgery plus adjuvant radiotherapy (45% vs. 63%, *p* = 0.01). Lastly, patients managed with active surveillance were less likely to have tumors greater than 60 mm compared to those treated with any other modality except surgery (37% vs. 45%, *p* = 0.25). 

### 3.6. Progression-Free Survival and Time to Next Treatment by Tumor Location

Given that DTs can show a different biological and clinical behavior based on their site of origin, we analyzed PFS and TTNT after first-line treatment in DTs arising at different locations (Figure 7). A detailed description of the frequency of each treatment used for DTs arising at different anatomical sites is provided in Appendix A.

In patients with DTs of the extremities, surgery plus radiotherapy was the only treatment associated with a reduced risk of progression in the multivariate analysis (HR: 0.3, 95%CI: 0.1–0.8, *p* = 0.01, Figure 7A, Appendix A). Additionally, in DTs of the extremities, surgery plus radiotherapy achieved longer 5-year PFS (63.0%, 95%CI: 35.4–81.4%) than either surgery alone (22.5%, 95%CI: 8.9–39.9%, *p* = 0.01), MR-HIFU/Cryoablation (0.0%, 95%CI: 0.0–0.0%, *p* < 0.0001), chemotherapy (16.7%, 95%CI: 0.9–51.7%, *p* = 0.001) or hormonal therapy (0.0%, 95%CI: 0.0–0.0%, *p* = 0.0001). However, there was no difference in 5-year PFS between patients with DTs of the extremities treated with surgery plus radiotherapy and those treated with TKIs (30.5%, 95%CI: 1.6–71.5%, *p* = 0.48) or those managed with active surveillance (41.7%, 95%CI: 5.7–7%, *p* = 0.25). Additionally, in DTs of the extremities, median TTNT was longer in patients treated with surgery plus radiotherapy (149.0 months, 95%CI: 54.9–243.0 months) compared to those who received either surgery alone (17.0 months, 95%CI: 5.9–28.0 months, *p* = 0.003), MR-HIFU/Cryoablation (9.0 months, 95%CI: 0.0–19.2 months, *p* < 0.0001), hormonal therapy (10.0 months, 95%CI: 0.3–19.6 months, *p* = 0.0005), chemotherapy (1.8 months, 95%CI: 0.3–19.6 months, *p* < 0.0001), or TKIs (20 months, 95%CI: 5.3–34.6 months, *p* = 0.04), but similar to those managed with active surveillance (33.0 months, 95%CI: 20.4–45.5 months, *p* = 0.44, Figure 7B). In the multivariate analysis, surgery plus radiotherapy was associated with longer TTNT (HR: 0.3, 95%CI: 0.1–0.7, *p* = 0.006), while chemotherapy correlated with shorter TTNT (HR: 2.7, 95%CI: 1.1–6.5, *p* = 0.02, Appendix A).

In patients with DTs of the abdominal wall, there was no difference in 5-year PFS between surgery (63.5%, 95%CI: 43.8–77.9%) and either MR-HIFU/Cryoablation (100%, *p* = 0.08) or active surveillance (0.0%, 95%CI: 0.0–0.0%, *p* = 0.18), while MR-HIFU/Cryoablation resulted in longer 5-year PFS than active surveillance (*p* = 0.008, Figure 7C, Appendix A). In DTs of the abdominal wall, surgery resulted in longer median TTNT (not reached) compared to hormonal therapy (3.0 months, 95%CI: not reached, *p* < 0.0001), while there was no difference in median TTNT between those treated with surgery and MR-HIFU/Cryoablation (not reached, *p* = 0.12), surgery and active surveillance (170.0 months, 95%CI: not reached, *p* = 0.75) or active surveillance and MR-HIFU/Cryoablation (*p* = 0.24, Figure 7D, Appendix A). Hormonal therapy was the only treatment associated with shorter TTNT in the multivariate analysis for DTs of the abdominal wall (HR: 14.9, 95%CI: 1.8–120.6, *p* = 0.01). 

For intra-abdominal DTs, there was no difference in 5-year PFS between surgery alone (68.2%, 95%CI: 49.5–81.1%) and either surgery plus radiotherapy (50.0%, 95%CI: 0.6–91.0%, *p* = 0.75), surgery plus systemic therapy (75.0%, 95%CI: 12.8–96.0%, *p* = 0.28), or chemotherapy (75.0%, 95%CI: 12.8–96.0%, *p* = 0.68, Figure 7E, Appendix A). Similarly, in patients with intra-abdominal DTs, there was no difference in median TTNT between those treated with surgery alone and either surgery plus radiotherapy (*p* = 0.39) or surgery plus systemic therapy (*p* = 0.28), while chemotherapy resulted in worse median TTNT (3.0 months, 95%CI: 0.0–25.0 months) than either surgery alone (not reached, *p* < 0.0001), surgery plus radiotherapy (*p* = 0.04), or surgery plus systemic therapy (68 months, 95%CI: 0.0–164.7 months, *p* = 0.01, Figure 7F, Appendix A).

For patients with DTs of the chest wall, there was no difference in 5-year PFS between surgery alone (47.4%, 95%CI: 24.0–68.2%) and either TKIs (50.0%, 95%CI: 5.2–84.4%, *p* = 0.87), MR-HIFU/Cryoablation (50.0%, 95%CI: 0.6–91.0%, *p* = 0.56), or active surveillance (50.0%, 95%CI: 5.2–84.4%, *p* = 0.83, Appendix A). Additionally, in DTs of the chest wall, surgery resulted in longer median TTNT (not reached) compared to MR-HIFU/Cryoablation (4.0 months, 95%CI: not reached, *p* = 0.0009), but similar median TTNT compared to TKIs (30.0 months, 95%CI: 0.0–71.9, *p* = 0.60) or active surveillance (14.0 months, 95%CI: 6.1–21.8, *p* = 0.10, Appendix A).

In DTs of the head and neck, there was no difference in 5-year PFS between patients managed with surgery alone (40.0%, 95%CI: 5.2–75.2%) and either TKIs (50%, 95%CI: 5.8–84.4%, *p* = 0.98), surgery plus radiotherapy (100%, *p* = 0.11), or active surveillance (100%, *p* = 0.50, Appendix A). Similarly, in DTs of the head and neck, surgery resulted in similar median TTNT (69.0 months, 95%CI: 0.0–146.9 months) as either TKIs (9.0 months, 95%CI: not reached, *p* = 0.55), surgery plus radiotherapy (*p* = 0.17) or active surveillance (*p* = 0.65, Appendix A).

### 3.7. Adverse Events and Complications

Adverse events and complications after chemotherapy, tyrosine kinase inhibitors, MR-HIFU/cryoablation, hormone therapy and active surveillance are reported in Table 4. 

Gastrointestinal symptoms, including diarrhea and nausea, were the most common adverse effects observed in patients treated with imatinib with or without adjuvant radiotherapy and were mostly grade 1 or 2. In patients treated with sorafenib there were two cases (33%) of Drug Reaction with Eosinophilia and Systemic Symptoms (DRESS) syndrome, which developed in both instances within a week of starting treatment [20]. Hormonal therapy was relatively well tolerated, with the most common adverse event reported being hot flashes (*n* = 3, 30%). Grade 2 vomiting was observed in 50% of those treated with methotrexate-vinblastine (*n* = 3), while the patient who received doxorubicin-dacarbazine was hospitalized for neutropenic fever a week after the initiation of treatment.

In patients treated with local ablation, mild to moderate pain at the tumor site was the most common complication both in those who received MR-HIFU (*n* = 4, 40%) and in those who received cryoablation (*n* = 2, 20%), and it usually resolved in a week after the procedure.

Of the 27 patients treated with surgery plus adjuvant radiotherapy, only one (3.7%) developed a secondary sarcoma with a median follow-up of 160 months (range, 20–280 months). Among the 11 patients treated with imatinib plus adjuvant radiotherapy, none developed a secondary malignancy with a median follow-up of 45 months (range, 34–107 months).

Among patients managed with active surveillance, six patients (32%) experienced worsening pain at the tumor site, while seven patients experienced progressive tumor growth (37%). Additionally, 37% of patients managed with initial active surveillance required subsequent treatment, which included TKIs (*n* = 5, 26%), MR-HIFU (*n* = 1, 5%), and hormonal therapy (*n* = 1, 5%). Median follow-up time for patients with active surveillance was 18 months (3–145 months).

## 4. Discussion 

Here, we retrospectively analyzed PFS and TTNT after first-line therapy in patients with newly diagnosed desmoid tumors that were managed with either surgery, systemic medical therapy, local ablation with cryoablation or MR-HIFU, or active surveillance.

We found that patients treated with first-line anti-estrogenic agents had worse PFS and TTNT than those who received either surgery alone or surgery plus adjuvant radiotherapy [21]. However, when analyzing outcomes based on tumor site, we found that in extremity DTs hormonal therapy achieved worse PFS and TTNT than those treated with surgery plus radiotherapy, but similar to those who received surgery alone. Hormonal therapy was also used for intra-abdominal and abdominal wall DTs, and, in these cases, it resulted in similar PFS and TTNT as surgery and surgery plus radiotherapy (intra-abdominal DTs), and similar PFS as surgery alone (abdominal wall DTs). Additionally, in our study, 80% (*n* = 8) of the patients treated with an anti-estrogenic agent received tamoxifen with or without NSAIDs, with the remaining patients receiving leuprolide with or without NSAIDs (*n* = 2, 20%). The 5-year PFS rate for patients treated with hormonal therapy in our study was 27%, which is comparable to observations from prior studies. For example, a study with tamoxifen–sulindac combination in the pediatric setting showed a 5-year PFS rate of approximately 30%, while another study with tamoxifen with or without NSAIDs in patients older than 19 years showed a 5-year PFS of 10% [22,23]. 

Similar results were observed for patients who received first-line cytotoxic chemotherapy, with shorter PFS and TTNT compared to those who received either surgery alone or surgery plus adjuvant radiotherapy. Upfront chemotherapy also correlated with shorter TTNT than surgery plus adjuvant systemic therapy. Chemotherapy was employed for both patients with extremity and intra-abdominal DTs. In extremity DTs, chemotherapy resulted in worse PFS and TTNT than surgery plus radiotherapy, but similar PFS to surgery alone or active surveillance. Instead, in intra-abdominal DTs, chemotherapy resulted in similar PFS and worse TTNT compared to surgery with or without radiotherapy or systemic therapy. Patients treated with cytotoxic chemotherapy in our study mostly received the combination of methotrexate and vinblastine (*n* = 6, 46%), showing a 5-year PFS rate of 24.9%, which is similar to a prior study where the 5-year PFS rate for patients treated with methotrexate and vinca alkaloids was 25–30% [24]. Overall, these results seem to suggest that a primary surgical approach with or without adjuvant radiotherapy can achieve longer PFS and longer TTNT than either anti-estrogenic agents or chemotherapy. However, we must take into consideration that patients who received chemotherapy or hormonal therapy had larger tumors that involved mostly the extremities and were significantly younger than those who received surgery, all features that correlate with worse outcomes. 

Patients who initially received a TKI, including either sorafenib, pazopanib or imatinib with or without radiotherapy had similar PFS compared to patients who were treated with surgery alone, and to those treated with surgery plus adjuvant radiotherapy. This was true both in extremity DTs and in DTs of the chest wall and of the head and neck, where TKIs were used. These findings are interesting since among all systemic medical therapies, TKIs were the only ones to be non-inferior to surgery plus adjuvant radiotherapy in terms of PFS. This might be secondary to the fact that more than half of patients who received TKIs were treated with imatinib plus adjuvant radiotherapy, an approach that we have been implementing to achieve better local control and faster symptom relief for patients who have unresectable tumors due to the involvement of neurovascular structures [25]. Similar results were observed in terms of TTNT, where TKIs were inferior only to surgery plus radiotherapy, but were the only treatment modality to achieve similar TTNT compared to surgery alone. The same was true for extremity and chest wall DTs but not for DTs of the head and neck, where TTNT was similar between TKIs and surgery plus radiotherapy. A possible explanation for these findings is that patients treated with TKIs were older than those in other treatment groups, and, as we have shown, a younger age at diagnosis correlates with worse TTNT and PFS. Additionally, most of the patients in our study receiving a TKI were treated with imatinib with or without adjuvant radiotherapy (*n* = 14, 66.7%). These patients had longer PFS compared to that shown in prior studies, which again might be explained by the use of adjuvant radiotherapy in our patient cohort. For example, in a study from Penel et al., the 2-year PFS rate for adult patients with progressive and unresectable DTs treated with imatinib was 55% compared to 85% in our study [26]. Similarly, a previous phase II multicentric trial by Chugh et al. in patients with unresectable DTs older than 10 years treated with imatinib showed a 1-year PFS rate of 66%, again lower than that observed in our study, where the 1-year PFS rate was 85% for patients treated with imatinib with or without radiotherapy [27]. The second most common TKI used in our study was sorafenib (*n* = 6, 28.6%). Patients treated with sorafenib showed a 2-year PFS rate of 80%, similar to that demonstrated in a previous work by Gounder et al., where the 2-year PFS rate was 81% in the sorafenib arm [28]. Regarding the toxicity profile of TKIs in patients with DTs, imatinib with or without radiotherapy was relatively well tolerated, with the most common side effects being mild to moderate nausea or diarrhea. The only serious adverse events observed were two cases of DRESS that developed in patients receiving sorafenib.

We also evaluated patients who initially received cryoablation or MR-HIFU as locally ablative strategies alternative to surgery and medical management. Local ablation is increasingly being used as a treatment modality in desmoid tumors and soft tissue sarcomas in general [29,30,31]. Our patients who received cryoablation showed a 1-year PFS rate of 90%, which is similar to that shown in prior studies. A phase II trial by Kurtz et al. in patients with refractory or symptomatic extra-abdominal DTs treated with cryoablation showed a 1-year PFS of 86% [16]. Additionally, a metanalysis on patients with extra-abdominal DTs undergoing cryoablation showed a pooled 1-year PFS rate of 84% [32]. Interestingly, in our study, first-line cryoablation and MR-HIFU were non-inferior in terms of PFS to either surgery alone or surgery followed by systemic therapy, while patients treated with surgery plus adjuvant radiotherapy showed longer PFS than those who received cryoablation or MR-HIFU. This may in part reflect that those patients treated with cryoablation or MR-HIFU tended to have larger tumors mostly involving the extremities (features that correlate with worse outcomes) compared to patients treated with a surgical approach. Additionally, when looking at PFS by tumor site, we found that local ablation resulted in similar PFS to surgery alone, both in DTs of the extremities, chest wall and abdominal wall, but worse PFS than surgery plus radiotherapy in extremity DTs. These results overall suggest that local ablation with either cryoablation or high intensity focused ultrasound can achieve similar benefits in terms of tumor growth control compared to surgery with or without adjuvant systemic therapy, especially for large tumors of the extremities, with the advantage of avoiding the morbidity associated with surgical resection. Indeed, MR-HIFU and cryoablation were particularly well tolerated, with pain at the ablation site being the most common adverse effect recorded. Conversely, the increased PFS observed with surgery plus adjuvant radiotherapy compared to MR-HIFU and cryoablation must be weighed against the small but significant risk of secondary malignancies related to radiotherapy. In terms of TTNT, cryoablation and MR-HIFU did perform similarly to surgery with adjuvant systemic therapies but showed shorter TTNT compared to both surgery alone and surgery with radiotherapy. By tumor site, cryoablation and MR-HIFU resulted in shorter TTNT than surgery plus radiotherapy in extremity DTs and shorter TTNT than surgery alone in chest wall DTs, with similar TTNT as surgery alone in both extremity and abdominal wall DTs. These results might be explained by the fact that many patients undergoing cryoablation or MR-HIFU often require multiple sequential treatments to achieve complete tumor eradication, which might falsely shorten the TTNT.

We also looked at patients who were initially managed with active surveillance. We found that primary active surveillance was non-inferior in terms of PFS and TTNT compared to surgery with or without adjuvant systemic therapy, although patients who received surgery plus adjuvant radiotherapy had longer PFS and TTNT than those that underwent active surveillance. Instead, when looking at outcomes by tumor location, we found that active surveillance achieved similar PFS and TTNT compared to surgery alone in extremity, chest wall, abdominal wall and head and neck DTs, as well as compared to surgery plus radiotherapy in extremity DTs. Additionally, only approximately one third of patients initially managed with active surveillance experienced worsening pain or progressive tumor growth, requiring either medical treatment or local therapy. These results overall suggest that initial active surveillance should be preferred, whenever possible, to a primary surgical approach, which is in line with current guidelines and available evidence [33,34,35]. Indeed, while surgery followed by adjuvant radiotherapy correlated with increased PFS and TTNT compared to active surveillance, the morbidity associated with surgery and the risk of secondary malignancies related to radiotherapy must be both taken into consideration. Additionally, approximately 50% of patients treated with surgery in our study had relapsed at the time of the last follow-up and this must be considered when deciding on primary surgery versus surveillance.

Additionally, in our study patients initially managed with active surveillance had similar PFS to those who received either TKIs, hormonal therapy, chemotherapy, or cryoablation and MR-HIFU as first-line, and this was true at each tumor location except for abdominal wall DTs where active surveillance resulted in worse PFS than MR-HIFU and cryoablation. TTNT was instead longer for patients undergoing initial active surveillance compared to first-line hormonal therapy, cytotoxic chemotherapy, or cryoablation and MR-HIFU, but similar to patients treated with TKIs. However, at each tumor site TTNT was similar between patients managed with active surveillance and those who received either TKIs, hormonal therapy, chemotherapy or cryoablation and MR-HIFU. Overall, these results might be explained by the fact that patients with pauci-symptomatic DTs and with a less aggressive phenotype might be more likely to be managed expectantly resulting in longer TTNT. Additionally, patients undergoing active surveillance in our study tended to have smaller tumors than those receiving medical therapies or cryoablation and MR-HIFU. Another possibility is that patients treated initially with cytotoxic chemotherapy or hormonal therapy might need to start a second line of treatment for reasons other than tumor progression, for example symptom control. 

Regarding clinical features at diagnosis that correlated with prognosis, we found that a tumor location involving the extremities, a younger age and a greater tumor size correlated with worse PFS and TTNT, similarly to that shown in prior studies [36].

Lastly, this study presents several limitations. First, the long period of observation with consequent changes in care standards over time generates heterogeneity within the groups related to differences between patients treated at different time points. The inclusion of pediatric patients could also increase the heterogeneity of our study groups. Additionally, due to the retrospective nature of this study, treatment choices were influenced by the baseline characteristics of the patients. Additionally, each treatment group compared had a relatively small sample size, which limits the power of our observations. Additionally, DTs involving different sites were included in each treatment group, which could also influence outcomes given the different biological and clinical behavior of DTs arising at different locations. Finally, this analysis included patients who were treated at a single institution that offers a broader range of treatment options (e.g., cryoablation and MR-HIFU) than is generally available, which can reduce the translatability of our results to the general population.

## 5. Conclusions

In summary, our data indicate that for patients with extremity DTs, surgery with adjuvant radiotherapy is the best active treatment modality, even though the risk of secondary malignancies can limit its effectiveness. In DTs of the trunk, including both abdominal and chest wall, MR-HIFU and cryoablation performed similarly to surgery, thus representing a valid alternative for DTs arising at these locations. For intra-abdominal DTs surgery was the most common treatment modality, even though our results do not provide enough evidence to determine if medical therapy alone or the addition of adjuvant radiotherapy or systemic therapy to surgery could improve outcomes in patients with intra-abdominal DTs. 

Our results overall suggest that an initial approach of active surveillance should be considered for patients with smaller and minimally symptomatic DTs, while indicating that TKIs, local ablation, and surgery achieve similar outcomes in patients with more aggressive disease. Larger randomized prospective studies comparing specific treatment modalities will be needed in the future to validate our observations. 

## Figures and Tables

**Figure 1 cancers-14-03907-f001:**
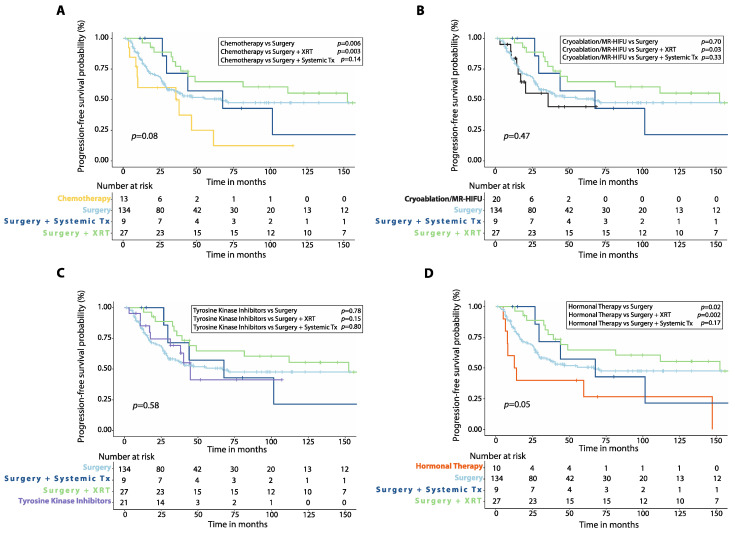
Progression-free survival (PFS) after first-line treatment in patients with desmoid tumors. Kaplan–Meier curves show comparison of PFS between patients treated with first-line chemotherapy (**A**), cryoablation or MR-HIFU (**B**), tyrosine kinase inhibitors (**C**), or hormonal therapy (**D**) versus surgery alone, surgery plus adjuvant radiotherapy (XRT) or surgery plus adjuvant systemic therapy. *p*-values were calculated with the Log-Rank method. XRT: radiotherapy; Systemic Tx: Systemic therapy; MR-HIFU: Magnetic resonance-guided high intensity focused ultrasound; Surgery plus systemic therapy includes surgery plus adjuvant chemotherapy, surgery plus adjuvant hormonal therapy and surgery plus imatinib.

**Figure 2 cancers-14-03907-f002:**
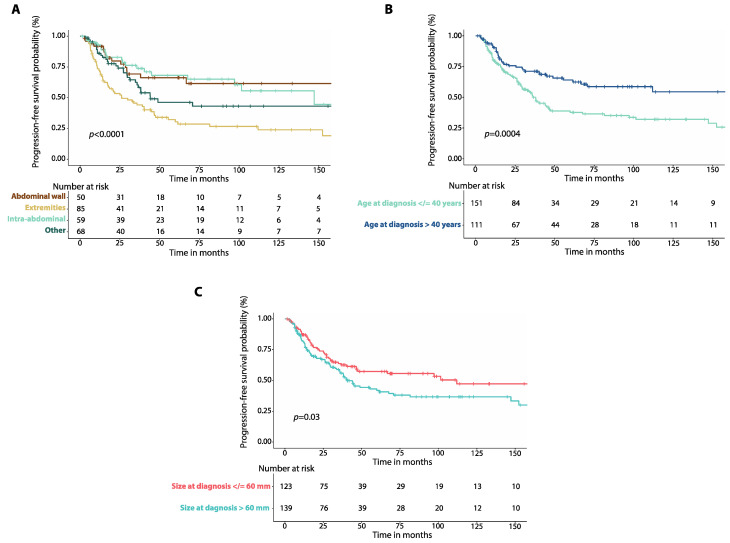
(**A**) Kaplan–Meier curves show progression-free survival (PFS) in patients with intra-abdominal desmoid tumors (DTs) compared to those with DTs of the extremities, DTs of the abdominal wall and DTs affecting other locations. (**B**) Kaplan–Meier curves show progression-free survival (PFS) in patients with DTs and age at diagnosis equal to or lower than 40 years compared to those with age at diagnosis greater than 40 years. (**C**) Kaplan–Meier curves show progression-free survival (PFS) in patients with DTs of size at diagnosis equal to or smaller than 60 mm compared to patients with tumor size greater than 60 mm. *p*-values were calculated with the Log-Rank method.

**Figure 3 cancers-14-03907-f003:**
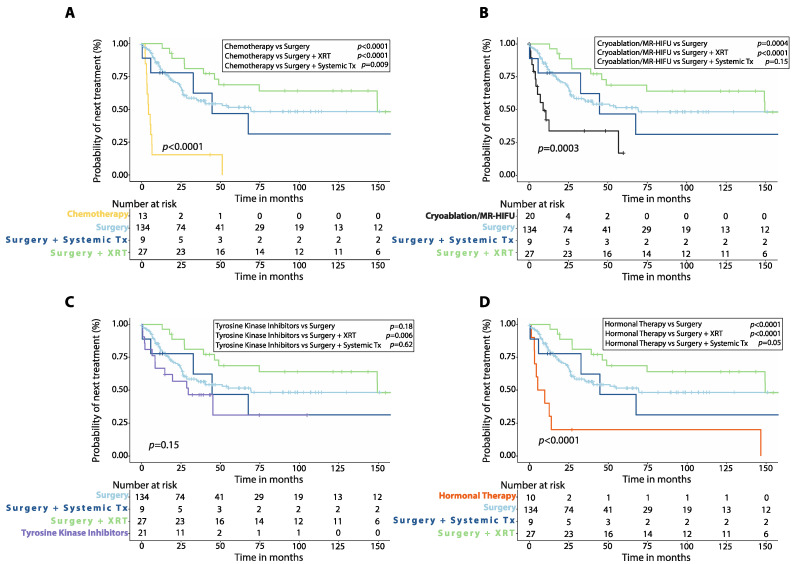
Time to next treatment (TTNT) after first-line therapy in patients with desmoid tumors. Kaplan–Meier curves show comparison of TTNT between patients treated with first-line chemotherapy (**A**), cryoablation or MR-HIFU (**B**), tyrosine kinase inhibitors (**C**), or hormonal therapy (**D**) versus surgery alone, surgery plus adjuvant radiotherapy (XRT) or surgery plus adjuvant systemic therapy. P-values were calculated with the Log-Rank method. XRT: radiotherapy; Systemic Tx: Systemic therapy; MR-HIFU: Magnetic resonance-guided high intensity focused ultrasound; Surgery plus systemic therapy includes surgery plus adjuvant chemotherapy, surgery plus adjuvant hormonal therapy, surgery plus imatinib.

**Figure 4 cancers-14-03907-f004:**
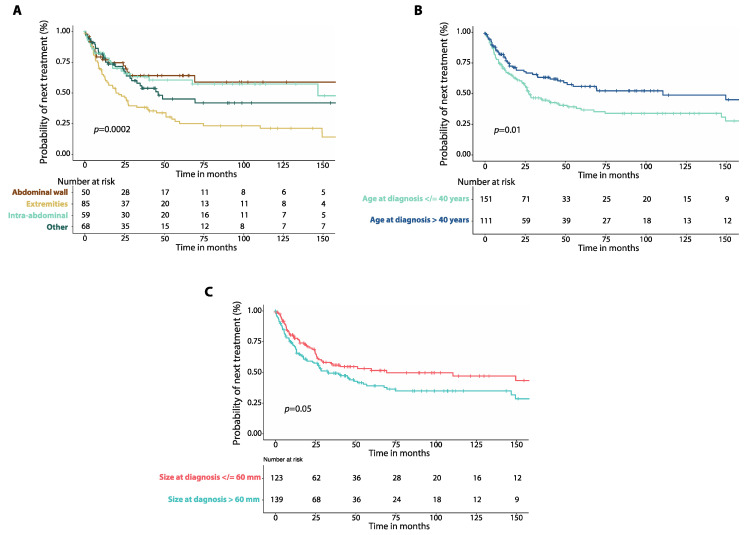
(**A**) Kaplan–Meier curves show time to next treatment (TTNT) after first-line therapy in patients with intra-abdominal desmoid tumors (DTs) compared to those of the extremities, of the abdominal wall and those affecting other locations. (**B**) Kaplan–Meier curves show time to next treatment (TTNT) in patients with DTs and age at diagnosis equal to or lower than 40 years compared to those with age at diagnosis greater than 40 years. (**C**) Kaplan–Meier curves show time to next treatment (TTNT) in patients with DTs and size of tumor at diagnosis equal to or smaller than 60 mm compared to patients with tumor size greater than 60 mm. *p*-values were calculated with the Log-Rank method.

**Figure 5 cancers-14-03907-f005:**
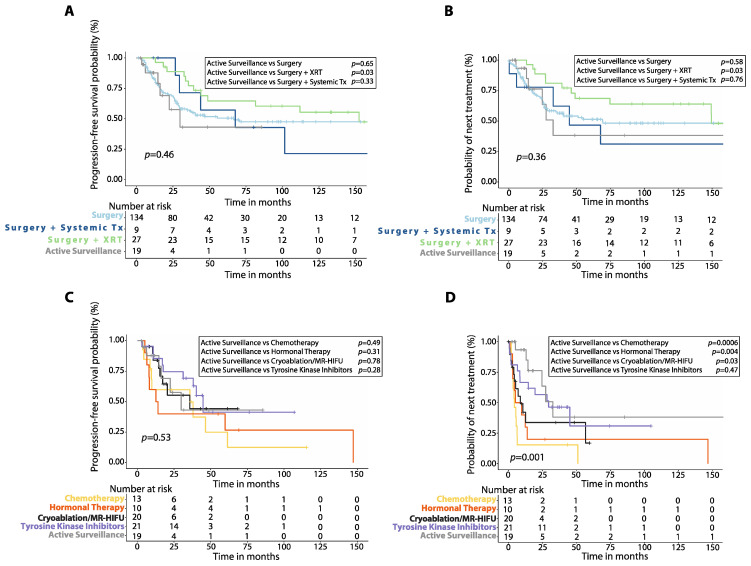
Progression-free survival (PFS) and time to next treatment (TTNT) after first-line therapy in patients with desmoid tumors managed with active surveillance versus other treatments. (**A**) Kaplan–Meier curve shows PFS of patients managed with active surveillance compared to those treated with upfront surgery, surgery plus adjuvant radiotherapy (XRT) or surgery plus adjuvant systemic therapy. (**B**) Kaplan–Meier curve shows TTNT after first-line therapy of patients managed with active surveillance compared to those treated with upfront surgery, surgery plus adjuvant radiotherapy (XRT) or surgery plus adjuvant systemic therapy. (**C**) Kaplan–Meier curve shows PFS of patients managed with active surveillance compared to those treated with first-line chemotherapy, hormonal therapy, cryoablation/MR-HIFU or tyrosine kinase inhibitors (TKIs). (**D**) Kaplan–Meier curve shows TTNT after first-line therapy of patients managed with active surveillance compared to those treated with first-line chemotherapy, hormonal therapy, cryoablation/MR-HIFU or TKIs. *p*-values were calculated with the Log-Rank method. XRT: radiotherapy; Systemic Tx: Systemic therapy; Surgery plus systemic therapy includes surgery plus adjuvant chemotherapy, surgery plus adjuvant hormonal therapy, surgery plus imatinib; MR-HIFU: Magnetic resonance-guided high intensity focused ultrasound.

**Figure 6 cancers-14-03907-f006:**
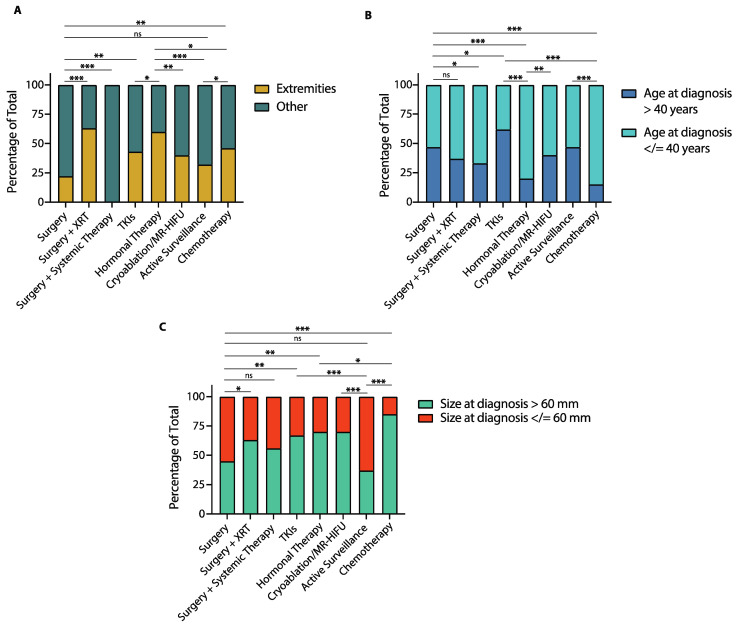
Tumor location, age at diagnosis and tumor size by treatment group in patients with desmoid tumors (DTs). (**A**) Bar graph shows the proportion of patients with DTs of the extremities compared to DTs involving other locations in each treatment group. (**B**) Bar graph shows the proportion of patients older than 40 years and 40 years or younger at diagnosis in each treatment group. (**C**) Bar graph shows the proportion of patients with tumors greater than 60 mm and 60 mm or smaller at diagnosis in each treatment group. XRT: radiotherapy; MR-HIFU: Magnetic resonance-guided high intensity focused ultrasound; TKIs: Tyrosine kinase inhibitors. (* *p* < 0.05; ** *p* < 0.01; *** *p* < 0.001; ns: non-statistically significant, Chi-square test).

**Figure 7 cancers-14-03907-f007:**
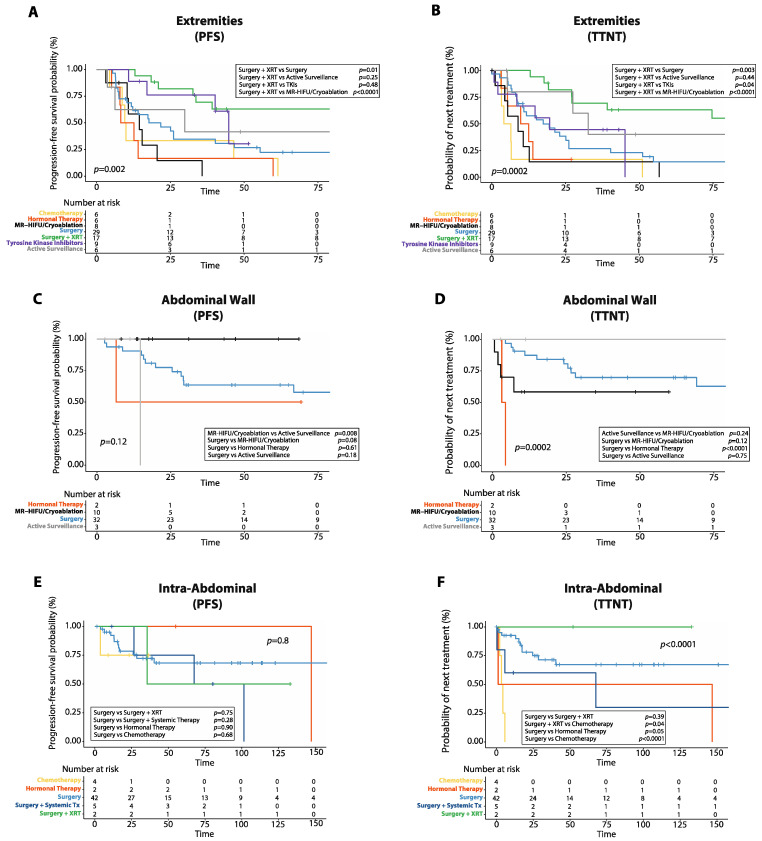
Progression-free survival (PFS) and time to next treatment (TTNT) after first line treatment in desmoid tumors (DTs) arising at different anatomical sites. (**A**) Kaplan–Meier curve shows PFS after first-line treatment in DTs of the extremities. (**B**) Kaplan–Meier curve shows TTNT after first-line treatment in DTs of the extremities. (**C**) Kaplan–Meier curve shows PFS after first-line treatment in DTs of the abdominal wall. (**D**) Kaplan–Meier curve shows TTNT after first-line treatment in DTs of the abdominal wall. (**E**) Kaplan–Meier curve shows PFS after first-line treatment in intra-abdominal DTs. (**F**) Kaplan–Meier curve shows TTNT after first-line treatment in intra-abdominal DTs. XRT: radiotherapy; MR-HIFU: magnetic resonance-guided high intensity focused ultrasound; Systemic Tx: systemic therapy.

**Table 1 cancers-14-03907-t001:** Baseline patient characteristics and treatment information.

**Age at Diagnosis**	
Median in years (range)	36.5 (0–87)
**Site**	
Extremities	85 (32.4%)
Intra-abdominal	59 (22.5%)
Abdominal wall	50 (19.0%)
Other *^a^*	68 (25.9%)
**Size at diagnosis (cm)**	
Median (range)	6.6 (1–30.0)
**Sex**	
Male (*n*, %)	77 (29.4%)
Female (*n*, %)	185 (70.6%)
**Treatment (First-line)**	
Surgery (*n*, %)	134 (51.1%)
Surgery + XRT (*n*, %)	27 (10.3%)
Surgery + Systemic Treatment (*n*, %)	9 (3.4%)
Chemotherapy (*n*, %)	13 (4.9%)
Hormonal Therapy (*n*, %)	10 (3.8%)
TKIs (*n*, %)	21 (8.0%)
Cryoablation/MR-HIFU (*n*, %)	20 (7.6%)
Active surveillance (*n*, %)	19 (7.2%)
Other *^b^* (*n*, %)	9 (3.4%)
**Follow-up (months)**	
Median (range)	63 (1–286)
**FAP**	
Yes (*n*, %)	14 (5.3%)
No (*n*, %)	248 (94.7%)
**Mutation**	
*CTNNB1* p.T41A (*n*, %)	20 (7.6%)
*CTNNB1* p.S45F (*n*, %)	8 (3.0%)
*APC* (*n*, %)	14 (5.3%)
Other *^c^* (*n*, %)	3 (1.1%)
Not available (*n*, %)	228 (87.0%)
**Status at last follow-up**	
Deceased (*n*, %)	4 (1.6%)
Alive WD (*n*, %)	118 (45.0%)
Alive NED (*n*, %)	140 (53.4%)

XRT: Radiotherapy; TKIs: Tyrosine-Kinase Inhibitors; MR-HIFU: Magnetic Resonance-guided high intensity focused ultrasound; WD: with disease; NED: no evidence of disease; Other *^a^*: Head and Neck (*n* = 13), Chest wall (*n* = 37), retroperitoneum (*n* = 2), pelvis (*n* = 3), scapula (*n* = 2), para-spinal (*n* = 9), intra-thoracic (*n* = 2); Other *^b^*: Radiotherapy only (*n* = 2), Nirocagestat (*n* = 1), Celecoxib (*n* = 4), Sulindac (*n* = 2). Other *^c^*: *CTNNB1* p.S45P (*n* = 1), other *CTNNB1* codon 45 mutations (*n* = 2).

**Table 2 cancers-14-03907-t002:** Progression-free survival after first-line treatment.

		PFS		Univariate Analysis		Multivariate Analysis	
Variables	Patients (Events)	1-Year (95% CI)	5-Year (95% CI)	HR (95% CI)	*p*-Value	HR (95% CI)	*p*-Value
**First-line Treatment**							
Surgery	134 (58)	82.2% (74.4–87.8%)	50.6% (41.0–59.5%)	0.9 (0.6–1.3)	0.66	1.5 (0.7–3.1)	0.26
Surgery plus XRT	27 (10)	100% (N/A)	64.9% (43.2–80%)	0.4 (0.2–0.9)	**0.02**	0.4 (0.1–0.9)	**0.04**
Surgery plus Systemic Therapy	9 (5)	100% (N/A)	57.1% (17.2–83.7%)	1.1 (0.4–2.5)	0.98	1.5 (0.5–5.0)	0.45
Chemotherapy	13 (9)	59.8% (28.5–80.0%)	24.9% (4.3–54.2%)	1.9 (1.0–3.9)	0.05	1.3 (0.5–3.4)	0.59
Hormonal Therapy	10 (8)	60.0% (25.3–82.7%)	26.7% (4.7–56.3%)	2.2 (1.1–4.5)	**0.03**	1.7 (0.7–4.7)	0.25
TKIs	21 (9)	85.2% (60.8–95%)	41.3% (13.7–67.4%)	0.9 (0.5–1.8)	0.83	0.9 (0.3–2.4)	0.83
Cryoablation/MR-HIFU	20 (8)	83.8% (57.7–94.4%)	44.4% (16.8–69.1%)	1.2 (0.6–2.5)	0.59	1.3 (0.5–3.5)	0.60
Active Surveillance	19 (6)	87.7% (58.8–96.8%)	43.1% (12.2–71.4%)	1.3 (0.5–2.8)	0.57	1.1 (0.4–3.1)	0.88
Other *^a^*	9 (5)	88.8% (43.2–98.3%)	55.5% (20.4–80.4%)	0.9 (0.4–2.2)	0.79	–	–
**Age (years)**							
>40	111 (37)	90.4% (82.9–94.7%)	64.1% (53.1–73.2%)	0.5 (0.4–0.7)	**<0.001**	–	–
≤40	151 (85)	79.7% (72.3–85.4%)	37.7% (28.8–46.5%)	1.9 (1.3–2.9)	**<0.001**	1.9 (1.3–2.9)	**0.002**
**Sex**							
Male	77 (38)	86.4% (76.3–92.4%)	53.3% (40.4–64.7%)	1.0 (0.7–1.5)	0.86	–	–
Female	185 (84)	83.1% (76.8–87.8%)	46.9% (38.4–54.9%)	0.9 (0.7–1.4)	0.86	0.7 (0.5–1.1)	0.18
**Tumor site**							
Intra-abdominal	59 (19)	92.8% (82.0–97.2%)	68.1% (52.2–79.7%)	0.5 (0.3–0.9)	**0.01**	0.4 (0.2–0.7)	**0.004**
Abdominal Wall	50 (15)	91.8% (79.6–96.8%)	66.2% (49.4–78.6%)	0.5 (0.3–0.9)	**0.03**	0.5 (0.3–0.9)	**0.04**
Extremities	85 (58)	72.3% (61.4–80.6%)	30.4% (20.2–41.2%)	2.2 (1.6–3.2)	**<0.001**	2.0 (1.2–3.2)	**0.003**
Other *^b^*	68 (30)	86.0% (74.9–92.4%)	46.2% (31.8–59.4%)	0.9 (0.6–1.4)	0.72	–	–
**Tumor size**							
>60 mm	139 (74)	84.3% (76.9–89.5%)	42.1% (32.6–51.3%)	1.5 (1.0–2.1)	**0.04**	1.4 (0.9–2.1)	0.06
≤60 mm	123 (48)	83.8% (75.8–89.3%)	57.2% (46.7–66.4%)	0.7 (0.5–0.9)	**0.04**	–	–
**Mutation**							
*CTNNB1* p.T41A	20 (9)	89.5% (65.1–97.3%)	65.4% (38.3–82.9%)	1.4 (0.7–2.8)	0.28	1.8 (0.9–3.7)	0.11
*CTNNB1* p.S45F	8 (4)	85.7% (33.4–97.8%)	85.7% (33.4–97.8%)	1.8 (0.7–4.9)	0.24	1.5 (0.5–4.5)	0.41
*APC*	14 (9)	92.9% (59.1–99.0%)	50.6% (21.2–74.1%)	1.2 (0.6–2.3)	0.62	1.7 (0.7–3.7)	0.20
Other *^c^*	3 (3)	66.7% (5.4–94.5%)	33.3% (0.9–77.4%)	1.8 (0.6–5.6)	0.32	2.2 (0.6–8.0)	0.21

XRT: Radiotherapy; TKIs: Tyrosine-Kinase Inhibitors; MR-HIFU: Magnetic Resonance-guided high intensity focused ultrasound; Other *^a^*: Head and Neck (*n* = 13), Chest wall (*n* = 37), retroperitoneum (*n* = 2), pelvis (*n* = 3), scapula (*n* = 2), para-spinal (*n* = 9), intra-thoracic (*n* = 2); Other *^b^*: Radiotherapy only (*n* = 2), Nirocagestat (*n* = 1), Celecoxib (*n* = 4), Sulindac (*n* = 2). Other *^c^*: *CTNNB1* p.S45P (*n* = 1), other *CTNNB1* codon 45 mutations (*n* = 2). The following variables were excluded from the multivariate analysis: Other treatment, Age > 40 years at diagnosis, male sex, other tumor site, tumor size ≤ 60 mm. Significant *p*-values are indicated in bold.

**Table 3 cancers-14-03907-t003:** Time to next treatment after first-line therapy.

		TTNT		Univariate Analysis		Multivariate Analysis	
Variables	Patients (Events)	Median (Months)	(95% CI)	HR (95% CI)	*p* Value	HR (95% CI)	*p* Value
**First-line Treatment**							
Surgery	134 (58)	69.1	Not reached	0.6 (0.4–0.9)	**0.008**	0.8 (0.4–1.6)	0.54
Surgery plus XRT	27 (11)	149.5	Not reached	0.4 (0.2–0.7)	**0.005**	0.3 (0.1–0.7)	**0.004**
Surgery plus Systemic Therapy	9 (5)	44.7	2.9–86.4	1.0 (0.4–2.5)	0.95	0.9 (0.3–3.0)	0.97
Chemotherapy	13 (12)	4.4	2.0–6.7	4.9 (2.8–9.2)	**<0.001**	4.0 (1.7–9.6)	**0.002**
Hormonal Therapy	10 (9)	5.3	0.0–13.6	3.3 (1.7–6.6)	**<0.001**	1.8 (0.7–4.5)	0.21
TKIs	21 (12)	29.6	8.0–51.1	1.3 (0.7–2.3)	0.39	0.9 (0.4–2.3)	0.98
Cryoablation/MR-HIFU	20 (12)	8.9	2.7–15.1	2.4 (1.3–4.4)	**0.004**	2.2 (0.9–5.4)	0.07
Active Surveillance	19 (7)	32.7	22.4–42.9	0.9 (0.4–1.9)	0.81	0.6 (0.2–1.7)	0.40
Other *^a^*	9 (7)	45.1	25.8–64.5	0.8 (0.5–1.2)	0.38	–	–
**Age (years)**							
>40	111 (46)	110.5	25.3–195.6	0.6 (0.4–0.9)	**0.01**	–	–
≤40	151 (87)	27.5	19.2–35.7	1.6 (1.1–2.2)	**0.01**	1.5 (0.9–2.2)	0.05
**Sex**							
Male	77 (38)	67.6	40.4–94.8	0.9 (0.6–1.2)	0.48	–	–
Female	185 (95)	34.8	17.9–51.7	1.1 (0.8–1.7)	0.48	1.1 (0.7–1.7)	0.61
**Tumor site**							
Intra-abdominal	59 (22)	146.7	Not reached	0.6 (0.4–1.0)	0.06	0.6 (0.3–1.2)	0.16
Abdominal Wall	50 (18)	170.1	Not reached	0.6 (0.4–1.0)	0.05	0.6 (0.3–1.1)	0.10
Extremities	85 (62)	19.6	10.4–28.8	2.1 (1.5–2.9)	**<0.001**	1.6 (0.9–2.6)	0.05
Other *^b^*	68 (31)	46.1	11.7–80.5	0.8 (0.6–1.2)	0.38	–	–
**Tumor size**							
>60 mm	139 (79)	32.7	17.7–47.8	1.4 (0.9–1.9)	0.06	1.1 (0.7–1.5)	0.74
≤60 mm	123 (54)	69.1	0.0–163.8	0.7 (0.5–1.0)	0.06	–	–
**Mutation**							
*CTNNB1* p.T41A	20 (11)	17.4	4.0–30.8	1.6 (0.9–3.0)	0.12	1.6 (0.8–3.2)	0.14
*CTNNB1* p.S45F	8 (6)	11.8	0.0–23.8	2.5 (1.1–5.8)	0.02	2.1 (0.9–5.0)	0.09
*APC*	14 (9)	35.8	15.3–56.3	1.2 (0.6–2.4)	0.55	1.6 (0.8–3.5)	0.20
Other *^c^*	3 (2)	24.6	21.7–27.5	1.2 (0.3–5.0)	0.75	0.5 (0.1–2.1)	0.32

XRT: Radiotherapy; TKIs: Tyrosine-Kinase Inhibitors; MR-HIFU: Magnetic Resonance-guided high intensity focused ultrasound; Other *^a^*: Head and Neck (*n* = 13), Chest wall (*n* = 37), retroperitoneum (*n* = 2), pelvis (*n* = 3), scapula (*n* = 2), para-spinal (*n* = 9), intra-thoracic (*n* = 2); Other *^b^*: Radiotherapy only (*n* = 2), Nirocagestat (*n* = 1), Celecoxib (*n* = 4), Sulindac (*n* = 2). Other *^c^*: *CTNNB1* p.S45P (*n* = 1), other *CTNNB1* codon 45 mutations (*n* = 2); The following variables were excluded from the multivariate analysis: Other treatment, Age > 40 years at diagnosis, male sex, other tumor site, tumor size ≤ 60 mm. Significant *p*-values are indicated in bold.

**Table 4 cancers-14-03907-t004:** Adverse effects and complications after first-line therapy in desmoid tumors.

Treatment Modality	Adverse Effect and Complications (*n*, %)	*n* Grade 1 Toxicities	*n* Grade 2 Toxicities	*n* Grade 3 Toxicities
Imatinib (*n* = 5)				
	Diarrhea (2, 40%)	1	1	0
	fatigue (1, 20%)	0	1	0
	Lower extremity edema (1, 20%)	1	0	0
Imatinib + XRT (*n* = 9)				
	Diarrhea (2, 22%)	1	1	0
	Nausea (2, 22%)	1	1	0
	Lower extremity edema (1, 11%)	1	0	0
	Dermatitis (1, 11%)	0	1	0
Sorafenib (*n* = 6)				
	Dermatitis (1, 17%)	0	1	0
	DRESS (2, 33%)			
Pazopanib (*n* = 1)				
	Hypertension (1, 100%)	0	1	0
Hormonal Therapy				
	Hot flashes (3, 30%)	2	1	0
Methotrexate-Vinblastine (*n* = 6)				
	Vomiting (3, 50%)	0	2	0
Doxorubicin (*n* = 4)				
	Hair loss (1, 25%)	0	1	0
Doxorubicin-Dacarbazine (*n* = 1)				
	Neutropenic fever (1, 100%)	0	0	1
MR-HIFU (*n* = 10)				
	Pain at the ablation site (4, 40%)	3	1	0
Cryoablation (*n* = 10)				
	Pain at the ablation site (2, 20%)	2	0	0
Active Surveillance (*n* = 19)				
	Tumor pain (6, 32%)	0	3	0
	Tumor growth (6, 32%)			

MR-HIFU: magnetic resonance-guided high intensity focused ultrasound; XRT: radiotherapy.

## Data Availability

The data and supporting findings of this study are available from the corresponding author, S.T., upon reasonable request.

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
