# Peer review of "Management of Patients with Newly Diagnosed Desmoid Tumors in a First-Line Setting"

_cancers, 2022, doi:10.3390/cancers14163907_

Round 1

Reviewer 1 Report

1. In the case of desmoid tumors, the clinical behavior differs depending on the tumor site, and therefore, the therapeutic strategy also differs. For example, when the site is Intra-abdominal, drug therapy or active surveillance is rarely chosen and resection is selected for histological diagnosis. When the site is Abdominal Wall, simple resection is performed, which rarely leads to postoperative recurrence. However, when the site is Extremities, recurrence often occurs even if R0 resection can be performed. This paper actually shows a significant difference in PFS between the Intra-abdominal, Abdominal Wall and Extremities Groups. In addition, when the site is Neck in the Other Group, the tumor often extends to the cervical and/or brachial plexus and R0 resection is difficult in many cases, which results in a high rate of postoperative recurrence. On the other hand, in most cases of Chest Wall in the Other Group, R0 resection can be performed and the postoperative recurrence rate is low. Therefore, it seems inappropriate to treat all cases with different tumor growth patterns together in this paper. The results of treatment should be examined separately by tumor site. The conclusion should indicate the treatment strategy for each site. 2. If this paper aims to clarify what should be selected as the initial treatment, complications from the treatment should also be clarified: for example, adverse effects of long-term administration of TKIs to benign, desmoid tumors, or complications due to combined RT. Also, it is necessary to mention the adverse effects of delayed initiation of treatment by performing active surveillance.

Author Response

Dear Reviewer,

Thank you for your thoughtful comments and review of our manuscript.

Please find below a detailed reply to your comments:

  1. In the case of desmoid tumors, the clinical behavior differs depending on the tumor site, and therefore, the therapeutic strategy also differs. For example, when the site is Intra-abdominal, drug therapy or active surveillance is rarely chosen and resection is selected for histological diagnosis. When the site is Abdominal Wall, simple resection is performed, which rarely leads to postoperative recurrence. However, when the site is Extremities, recurrence often occurs even if R0 resection can be performed. This paper actually shows a significant difference in PFS between the Intra-abdominal, Abdominal Wall and Extremities Groups. In addition, when the site is Neck in the Other Group, the tumor often extends to the cervical and/or brachial plexus and R0 resection is difficult in many cases, which results in a high rate of postoperative recurrence. On the other hand, in most cases of Chest Wall in the Other Group, R0 resection can be performed and the postoperative recurrence rate is low. Therefore, it seems inappropriate to treat all cases with different tumor growth patterns together in this paper. The results of treatment should be examined separately by tumor site. The conclusion should indicate the treatment strategy for each site.
    • We decided to analyze patients by treatment modality without doing a separate analysis based on tumor location given that in this case we would compare very small groups where often the sample size is less than 10. Also, tumor site is included in the multivariate analysis for both PFS and TTNT to try to control for the effects of tumor location on outcomes. We have also made a note of this issue at the end of the discussion section. Additionally, we have provided a new image in the supplemental material where a more detailed breakdown of the primary tumor site in each treatment group is shown (Figure S7). Lastly, we compared PFS and TTNT between patients with desmoid tumors of the chest wall and those with desmoid tumors of the head and neck and we found no difference between the two groups (data added to Figure S4 and Figure S6, respectively).
  2. If this paper aims to clarify what should be selected as the initial treatment, complications from the treatment should also be clarified: for example, adverse effects of long-term administration of TKIs to benign, desmoid tumors, or complications due to combined RT. Also, it is necessary to mention the adverse effects of delayed initiation of treatment by performing active surveillance.
    • Data regarding adverse effects has been included as requested. Information regarding adverse effects of surgery is not included as this information is available from historical controls in the literature.

Reviewer 2 Report

It is necessary to clearly state how long active surveillance is conducted. In addition, it is assumed that many cases in which surgery or other treatment are performed after active surveillance are included, but it should be stated more clearly whether these cases are included in the number of surgical cases or not.

Author Response

Dear Reviewer,

Thank you for your thoughtful comments and review of our manuscript.

Please find below a detailed reply to your comments:

  1. It is necessary to clearly state how long active surveillance is conducted.
    • We have included the following statement in the methods section to describe the modalities of active surveillance at our institution: “At our institution, active surveillance consisted in clinical visits and either CT or MRI imaging performed every three months during the first year after diagnosis, every six months during the second year, and once a year for the following years until disease progression.”
  2. In addition, it is assumed that many cases in which surgery or other treatment are performed after active surveillance are included, but it should be stated more clearly whether these cases are included in the number of surgical cases or not.
    • Only patients that were treated upfront with surgery or other treatments without undergoing active surveillance beforehand were included in the treatment groups. We have included the following statement in the methods section to clarify this point: “For the other treatment groups, including surgery, medical treatments, and local ablation, only patients that received these treatment modalities as first-line, without undergoing active surveillance beforehand, were included.”

Reviewer 3 Report

This is a moderate size retrospectic series for desmoid fibromatosis of 262 patients. The authors present a well developed analysis of treatment outcomes in the first line setting for desmoid fibromatosis. A multivariate analysis is conducted. However, with multiple different treatment catergoies, and only 262 patients, with differences in the standard treatments used over the 30 years study appear, potential differences in treatments of adult versus pediatric patients, dirver mutation, and selection bias in terms of treatment specifically those that may have required multiple treatments, surgery, radiation, and or chemo their multivariate analysis does not account for all of these different variables. It becomes very difficult to draw conclusions from this dataset. Additional, the decision for treatment (symptoms, pain, etc..) was not specifically mentioned, and the PFS was determined by symptoms or tumor enlargement (ie, not clinical trial response criteria). The authors attempt to analyze some of the factors that may led to selection bias, ei age, location, size, and mention this limitation in the discussion and conclusion. Still this is a severe limiation of this study.

Also, section 3.2 remove "put in same paragraph above"

Author Response

Dear Reviewer,

Thank you for your thoughtful comments and review of our manuscript.

Please find below a detailed reply to your comments:

This is a moderate size retrospectic series for desmoid fibromatosis of 262 patients. The authors present a well developed analysis of treatment outcomes in the first line setting for desmoid fibromatosis. A multivariate analysis is conducted. However, with multiple different treatment catergoies, and only 262 patients, with differences in the standard treatments used over the 30 years study appear, potential differences in treatments of adult versus pediatric patients, dirver mutation, and selection bias in terms of treatment specifically those that may have required multiple treatments, surgery, radiation, and or chemo their multivariate analysis does not account for all of these different variables. It becomes very difficult to draw conclusions from this dataset. Additional, the decision for treatment (symptoms, pain, etc..) was not specifically mentioned, and the PFS was determined by symptoms or tumor enlargement (ie, not clinical trial response criteria). The authors attempt to analyze some of the factors that may led to selection bias, ei age, location, size, and mention this limitation in the discussion and conclusion. Still this is a severe limiation of this study.

  • We have added a statement to describe the criteria used at our institution to decide which patients should undergo initial treatment rather than active surveillance (Materials and Methods section, paragraph 2.2, line 98: “The decision to pursue treatment rather than active surveillance was based on presence of symptoms and proximity to vital structures or organs with high risk of disability and morbidity in case of tumor progression”).
  • Also, we are aware of the limitations of our study that you point out and we have tried as much as possible to clearly state these limitations in the discussion section.
  • To address your other concerns:
    • We have not included the number of subsequent treatments received after first-line therapy in our multivariate analysis given that we focused specifically on outcomes that are not influenced by second line and subsequent therapies (i.e., PFS and TTNT after first-line therapy).
    • The number of pediatric patients is relatively limited (33 patients aged 18 years or younger) compared to the totality of the dataset (262 patients). However, we have made a note of this in the discussion.
    • We have included information regarding driver mutations in the multivariate analysis, but unfortunately tumor DNA testing was not available for many patients at diagnosis given that this test has been introduced relatively recently in clinical practice.

Round 2

Reviewer 1 Report

1. In the case of desmoid tumors, the clinical behavior differs depending on the tumor site, and therefore, the therapeutic strategy also differs. For example, when the site is Intra-abdominal, drug therapy or active surveillance is rarely chosen and resection is selected for histological diagnosis. When the site is Abdominal Wall, simple resection is performed, which rarely leads to postoperative recurrence. However, when the site is Extremities, recurrence often occurs even if R0 resection can be performed. This paper actually shows a significant difference in PFS between the Intra-abdominal, Abdominal Wall and Extremities Groups.

In addition, when the site is Neck in the Other Group, the tumor often extends to the cervical and/or brachial plexus and R0 resection is difficult in many cases, which results in a high rate of postoperative recurrence. On the other hand, in most cases of Chest Wall in the Other Group, R0 resection can be performed and the postoperative recurrence rate is low. Therefore, it seems inappropriate to treat all cases with different tumor growth patterns together in this paper. The results of treatment should be examined separately by tumor site. The conclusion should indicate the treatment strategy for each site.

2. If this paper aims to clarify what should be selected as the initial treatment, complications from the treatment should also be clarified: for example, adverse effects of long-term administration of TKIs to benign, desmoid tumors, or complications due to combined RT. Also, it is necessary to mention the adverse effects of delayed initiation of treatment by performing active surveillance.
